# T-square resistivity without Umklapp scattering in dilute metallic Bi$_2$O$_2$Se

Jialu Wang[1,2], Jing Wu[1,2], Tao Wang[1,2], Zhuokai Xu[1,2], Jifeng Wu[1,2], Wanghua Hu[1,2], Zhi Ren[1,2], Shi Liu [1,2], Kamran Behnia[3] & Xiao Lin [1,2✉]

Fermi liquids (FLs) display a quadratic temperature ($T$) dependent resistivity. This can be caused by electron-electron (e-e) scattering in presence of inter-band or Umklapp scattering. However, dilute metallic SrTiO$_3$ was found to display $T^2$ resistivity in absence of either of the two mechanisms. The presence of soft phonons as possible scattering centers raised the suspicion that $T^2$ resistivity is not due to e-e scattering. Here, we present the case of Bi$_2$O$_2$Se, a layered semiconductor with hard phonons, which becomes a dilute metal with a small single-component Fermi surface upon doping. It displays $T^2$ resistivity well below the degeneracy temperature in absence of Umklapp and inter-band scattering. We observe a universal scaling between the $T^2$ resistivity prefactor ($A$) and the Fermi energy ($E_F$), an extension of the Kadowaki-Woods plot to dilute metals. Our results imply the absence of a satisfactory understanding of the ubiquity of e-e $T^2$ resistivity in FLs.

[1] School of Science, Westlake University, 18 Shilongshan Road, 310024 Hangzhou, Zhejiang Province, China. [2] Institute of Natural Sciences, Westlake Institute for Advanced Study, 18 Shilongshan Road, 310024 Hangzhou, Zhejiang Province, China. [3] Laboratoire Physique et Etude de Matériaux (CNRS-Sorbonne Université-ESPCI Paris), PSL Research University, Paris 75005, France. ✉email: linxiao@westlake.edu.cn

Collision between electrons of a metal leads to a $T$-square resistivity. Postulated in 1930s by Landau and Pomeranchuk[1] and independently by Baber[2], this feature has been widely documented in elemental[3] and strongly correlated[4] metals. At sufficiently low temperature, their resistivity ($\rho$) follows this simple expression:

$$\rho = \rho_0 + AT^2 \qquad (1)$$

The residual resistivity, $\rho_0$, depends on defects but, $A$ is an intrinsic property of the metal, found to scale with the electronic-specific heat[3,4] in dense metals (i.e., those having roughly one carrier per formula unit). The quadratic temperature dependence of the phase space is a consequence of the fact that both participating electrons reside within a thermal window of the Fermi level.

In absence of a lattice, an electron–electron collision conserves momentum and cannot degrade the charge current. To generate finite resistivity, such collisions should transfer momentum to the lattice. There are two known mechanisms: either it is because there are multiple electron reservoirs unequally coupled to the lattice[2,3], or because the collision is an Umklapp event[5,6]. In the first case, the two colliding electrons have distinct electron masses[2]. Momentum transfer between these two distinct reservoirs sets the temperature dependence of resistivity, and the mass mismatch causes momentum leak to the lattice thermal bath. In the second case, one of the two colliding electrons is scattered to the second Brillouin zone and returns to the first one by transferring a unit vector of the reciprocal lattice ($\mathbf{G}$) of momentum to the lattice[5,6].

The observation of a $T^2$ resistivity in dilute metallic $SrTiO_3$ indicated, however, that our understanding of the microscopic foundations of this ubiquitous phenomenon is unsatisfactory[7]. $SrTiO_3$ is a cubic perovskite at room temperature. It is a quantum paraelectric and becomes a dilute metal upon introduction of a tiny concentration of mobile electrons[8]. Three concentric conducting bands centered at $\Gamma$ point of the Brillouin zone are successively filled[9]. The quadratic temperature dependence of its electrical resistivity[10,11] persists[7] even when its Fermi surface shrinks to a single pocket[12] and none of the two mechanisms operate. However, it was more recently suggested that this enigmatic $T$-square resistivity may be caused by exotic mechanisms such as scattering by magnetic impurities[13] or by two soft transverse optical phonons (See Supplementary Discussion)[14,15]. Such soft phonons are known to play a decisive role in transport properties of the system, at least at high temperatures[16–19].

Here, we report on another dilute metal, doped $Bi_2O_2Se$. We show that it displays $T^2$ resistivity whilst the Hall carrier density ($n$) changes by two orders of magnitude and in absence of interband and Umklapp scattering. Moreover, there are no soft phonons and the $T$-square resistivity is restricted to temperatures well below the degeneracy temperature. The e–e origin of this $T$-square resistivity is unambiguous. Comparing the evolution of $A$ with $n$ and $E_F$ in $Bi_2O_2Se$ and $SrTiO_3$, we uncover a universal scaling between $A$ and $E_F$ in dense and dilute Fermi liquids. Our results imply that a proper microscopic theory of the link between electron–electron scattering and $T$-square resistivity is still missing.

## Results

### First-principle calculations of $Bi_2O_2Se$ single crystal.
Stoichiometric $Bi_2O_2Se$ is a layered semiconductor with tetragonal crystal structure (anti-$ThCr_2Si_2$ phase) at room temperature[20], shown in Fig. 1a. Available $Bi_2O_2Se$ single-crystals are metallic with extremely mobile carriers[21–23]. The insulator is doped by unavoidable defects, such as Se or O vacancies and Se–Bi antisite defects[24,25].

According to density functional theory (DFT) calculations[24], the conduction band is centered at the $\Gamma$ point of the Brillouin zone, following a parabolic dispersion (Fig. 1b and Supplementary Fig. 1). This has been revealed by angle-resolved photoemission spectroscopy (ARPES) measurements[21,26]. The Fermi surface is an elongated ellipsoid (Fig. 1a) seen by quantum oscillations[26]. The comparison of crystal and band structure between $Bi_2O_2Se$ and $SrTiO_3$ is summarized in Supplementary Table 1.

Our DFT calculations of phonon spectrum for the tetragonal phase of $Bi_2O_2Se$ are in agreement with what was reported previously[27,28]. Figure 1c presents the phonon dispersion at ambient pressure. The absence of imaginary frequencies implies that the tetragonal phase is dynamically stable. Figure 1d–f shows the evolution of the calculated local potential and phonon frequencies with hydrostatic pressure. Clearly, the single-well local potential at ambient pressure is different from a quantum paraelectric, where the local potential is a shallow double-well and the ferroelectric phase is aborted due to quantum tunneling between two local minima of the well. In a quantum paraelectric, the soft mode is very sensitive to hydrostatic pressure. For example, in PbTe a well-known quantum paraelectric, the soft mode frequency almost triples by reducing the lattice constant ($a$) by 1%[29]. In contrast, all three optical modes change moderately by reducing $a$, in Fig. 1f. The lowest mode ($TO_1$) hardens only by 20% by a similar reduction of $a$. We also note that a large negative pressure ($\approx -5$ GPa) is required to make the system ferroelectric. These observations rule out the proximity to a ferroelectric instability, excluding the presence of soft phonons.

### Quadratic temperature dependence of resistivity.
Figure 2a shows the temperature dependence of resistivity $\rho(T)$ in our $Bi_2O_2Se$ samples with various $n$ (See Supplementary Fig. 2, Supplementary Fig. 3 and Supplementary Table 2 for more information). We note that our data (obtained for $n < 1 \times 10^{19}$ cm$^{-3}$) smoothly joins what was recently reported for $n = 1.1 \times 10^{19}$ cm$^{-3}$[22]. Upon cooling from room temperature to 1.8 K, resistivity decreases by two orders of magnitude, comparable to what is seen in doped quantum paraelectrics such as $SrTiO_3$, $KTaO_3$, and PbTe[16].

Panels b–e of Fig. 2 document the low-temperature quadratic temperature dependence of resistivity. It persists down to the lowest temperatures in all the samples. Equation (1) holds below a characteristic temperature (dubbed $T_{quad}$). As seen in Fig. 2f, $T_{quad}$ is an order of magnitude lower than $T_F$, the degeneracy temperature of the fermionic system (extracted from our study of quantum oscillations described below). The relevance of $T_{quad} \ll T_F$ inequality in $Bi_2O_2Se$ contrasts with what was seen in $SrTiO_3$[7]. Only well below the degeneracy temperature, one expects the phase space of the fermion-fermion scattering to be quadratic temperature. Therefore, one reason to question the attribution of $T$-square resistivity to e–e scattering, which was raised in the case of $SrTiO_3$, is absent.

Figure 2g presents the evolution of the $T^2$ prefactor ($A$) with $n$ for $Bi_2O_2Se$. For comparison, the relevant data for doped $SrTiO_3$ and $KTaO_3$ are also shown. In both cases, $A$ decreases with increasing $n$. While $A(n)$ is more fluctuating in $Bi_2O_2Se$ than in $SrTiO_3$, the two slopes ($\frac{n}{A}\frac{dA}{dn}$) are close to each other. At similar $n$, the magnitude of $A$ differs by more than one order of magnitude lower. We will see below that this arises because of the difference in the magnitude of the effective electron mass, which sets the Fermi energy at a given carrier density.

Thus, $Bi_2O_2Se$ is the second metallic system in which $T$-square resistivity is observed in presence of a single-component and small Fermi surface. The absence of multiple pockets leaves no place for interband scattering. The smallness of the Fermi surface excludes Umkalpp events (See below for quantitative discussion).

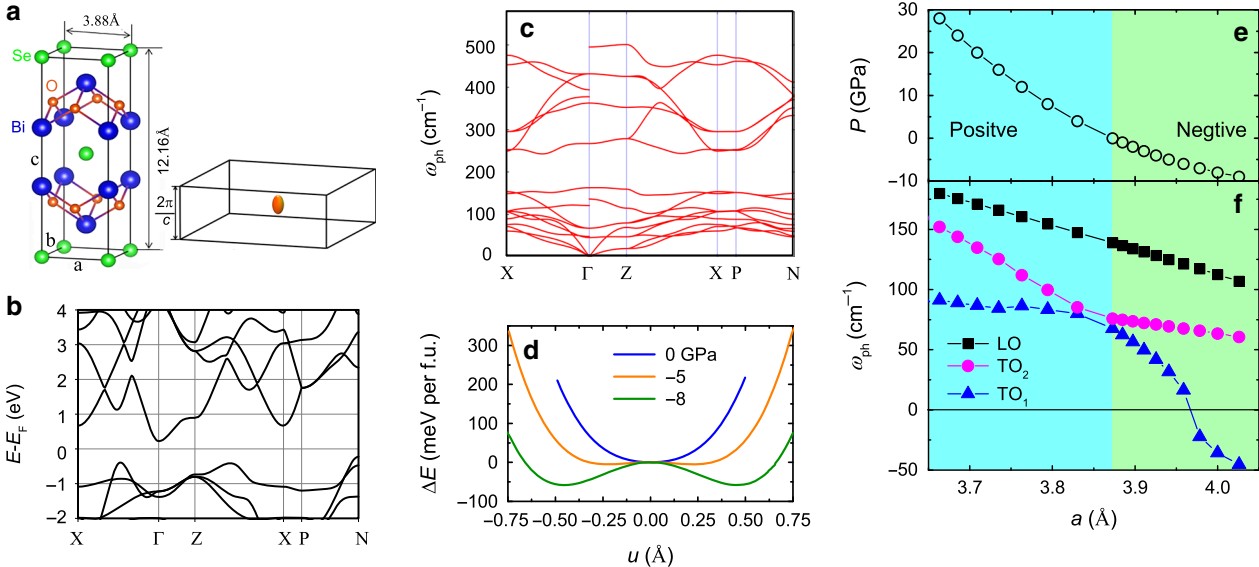

**Fig. 1 Electronic and phonon properties of Bi$_2$O$_2$Se from first-principle calculations. a** Unit cell of the tetragonal phase and sketch of the ellipsoid Fermi surface at the $\Gamma$ point of Brillouin zone. **b** Electronic band structures without spin-orbit coupling along high-symmetry lines of the Brillouin zone of the body-centered tetragonal unit cell. **c** Phonon dispersion at ambient pressure. **d** Local potential energy surface with respect to ion displacement along the lowest optical phonon mode, which changes from single-well to double-well by exerting large negative hydrostatic pressure. **e** One-to-one correspondence between the lattice parameter ($a$) and the hydrostatic pressure ($P$). **f** Frequency evolution of the three lowest optical modes with $a$, including two transverse modes: TO$_1$ (blue triangles) and TO$_2$ (magenta circles) and a longitudinal mode: LO (black squares) . The lattice constant at ambient pressure (3.873 Å) is taken to be the optimized value from DFT calculations and the experimental lattice parameter is 3.88 Å.

Two features make the theoretical challenge thrown down by Bi$_2$O$_2$Se more solid than the one presented by SrTiO$_3$. First of all, since $T_{\text{quad}} \ll T_{\text{F}}$, one objection to associating $T$-square resistivity and e–e scattering vanishes. Moreover, the absence of any exotic soft phonons, rules out their possible role as scattering centers[14,15].

Having shown that $T$-square resistivity is present in Bi$_2$O$_2$Se like in SrTiO$_3$ without objections ascribing it to e–e collisions, let us consider the relevance of the expression previously used[7] for the $T$-square prefactor:

$$A = \frac{\hbar}{e^2}\left(\frac{k_{\text{B}}}{E_{\text{F}}}\right)^2 l_{\text{quad}} \quad (2)$$

Here, $\hbar$ is Planck's constant divided by $2\pi$, e is the electron charge, $k_{\text{B}}$ is the Boltzmann constant and $l_{\text{quad}}$ is a material-dependent characteristic length, which arises uniquely due to a dimensional analysis. Such an examination requires to know $E_{\text{F}}$ at a given $n$. For this, we performed a detailed study of quantum oscillations.

**Resistive quantum oscillations.** Figure 3a, b shows that in presence of magnetic field, resistivity shows quantum oscillations with a single frequency for both in-plane and out-of-plane orientations of the magnetic field. When $n \approx 4.3 \times 10^{18}\,\text{cm}^{-3}$, the oscillation frequency is $F_{\text{H}\|c} \approx 51$ T and $F_{\text{H}\|\text{ab}} \approx 93$ T. This implies that the Fermi surface is an ellipsoid at the center of the Brillouin zone. The Fermi surface anisotropy $\alpha = \frac{F_{\text{H}\|\text{ab}}}{F_{\text{H}\|c}} = 1.8$. This is in excellent agreement with our theoretical calculations (Fig. 1b), which finds that the dispersion along $\Gamma$-X and $\Gamma$-Z with the value of anisotropy ($\alpha$) differ by 1.75. Similar data for other samples are presented in the Supplementary Fig. 4 and summarized in Supplementary Table 3.

The volume of the ellipsoidal Fermi pocket implies a carrier concentration $n_{\text{SdH}} = 3.8 \times 10^{18}\,\text{cm}^{-3}$, close to the Hall carrier

concentration. The numbers remain close to each other when $n > 10^{18}\,\text{cm}^{-3}$ (see Supplementary Table 3). At very low densities, $n_{\text{SdH}}$ starts to fall below $n$, which may indicate that the single Fermi sea begins to fall apart to isolated lakes due to insufficient homogeneity in doping.

The small size of the Fermi surface excludes the possible occurrence of Umklapp event, which requires a Fermi wave vector larger than one-fourth of the smallest reciprocal lattice vector, **G**. For Bi$_2$O$_2$Se, the smallest **G** is along $c$-axis, $\mathbf{G}_{\text{c}} = \frac{2\pi}{c} \approx 5.17\,\text{nm}^{-1}$, given the lattice constant $c = 12.16$ Å of the tetragonal unit cell. Then the threshold carrier density for Umklapp scattering ($n_{\text{U}}$) can be estimated to be $n_{\text{U}} = \frac{1}{3\pi^2}k_{\text{Fa}}^2 k_{\text{Fc}} = 3 \times 10^{19}\,\text{cm}^{-3}$ with $k_{\text{Fc}} = \frac{\mathbf{G}_{\text{c}}}{4}$ and $\alpha = \frac{k_{\text{Fc}}}{k_{\text{Fa}}} \approx 1.8$ ($k_{\text{Fc}}$ and $k_{\text{Fa}}$ are Fermi wave vector along $c$-axis and $a$-axis). For $n < n_{\text{U}}$, no Umklapp scattering is expected, which includes our range of study and beyond.

Figure 3c shows how quantum oscillations are damped with increasing temperature. This allows us to extract the effective mass $m^*$ using the Lifshitz–Kosevich (L–K) formula:

$$R_{\text{L}} = \frac{X}{\sinh(X)}, \quad X = \frac{\eta T m^*}{H} \quad (3)$$

where $\eta = \frac{2\pi^2 k_{\text{B}}}{e\hbar}$. We find an in-plane effective electron mass $m^*_{\text{H}\|c} \approx 0.17 m_{\text{e}}$, in agreement with previous reports[21,22,26], and an out-of-plane mass $m^*_{\text{H}\|\text{ab}} \approx 0.25 m_{\text{e}}$ ($m_{\text{e}}$ is the free electron mass). As expected, the ratio of these *cyclotron* masses is close to the ratio of the two ellipsoid axes.

Figure 3d shows $E_{\text{F}} = \frac{(\hbar k_{\text{Fab}})^2}{2m^*_{\text{H}\|c}}$ as a function of $k_{\text{Fab}}$ (the in-plane wave vector). Each data point represents a different sample. As seen in Fig. 3e, the magnitude of the effective mass and Fermi surface anisotropy remain unchanged with doping. The data imply that the conducting band dispersion is parabolic. We also note that the effective mass resolved here ($m^* = 0.17 \pm 0.1 m_{\text{e}}$, see Table S1) is only $1.3 \pm 0.1$ larger than the DFT calculated bare

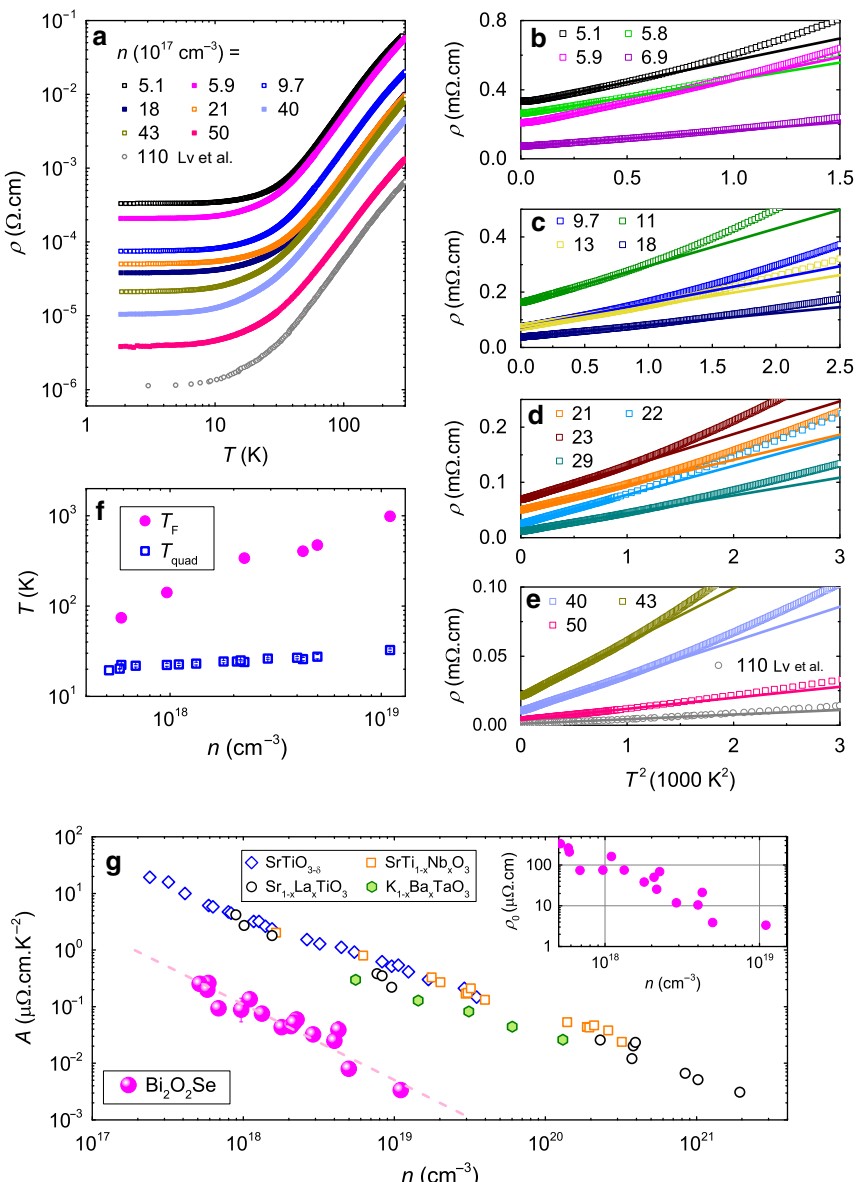

**Fig. 2 Temperature-dependent resistivity of Bi$_2$O$_2$Se at various Hall carrier concentrations ($n$). a** Resistivity as a function of temperature from 1.8–300K on a Log–Log scale. The data at $n \approx 1.1 \times 10^{19}$ cm$^{-3}$ are from ref. [22]. **b–e** The low-$T$ resistivity as a function of quadratic temperature. The linear lines are fits by Eq. (1). **f** $T_F$ and $T_{quad}$ as a function of $n$. $T_{quad}$ is a characteristic temperature above which the resistivity deviates from the $T^2$ behavior. $T_F$ and $T_{quad}$ are represented by solid magenta circles and open blue squares respectively. The error bars denote uncertainty in determining $T_{quad}$ from **b–e**. **g** The slope of $T^2$ resistivity ($A$) as a function of $n$ for Bi$_2$O$_2$Se: solid magenta circles, compared with doped SrTiO$_3$ (SrTiO$_{3-\delta}$[7]: open blue diamonds, SrTi$_{1-x}$Nb$_x$TiO$_3$[7,11]: open orange squares, and Sr$_{1-x}$La$_x$TiO$_3$[10]: open black circles) and K$_{1-x}$Ba$_x$TaO$_x$[45]: solid olive hexagons. The dashed line is a guide to eyes. The inset shows the variation of residual resistivity with increasing $n$ for Bi$_2$O$_2$Se.

mass of $m_b = 0.125 m_e$. In contrast, in SrTiO$_3$, band dispersion of the lower band is nonparabolic and there is a significant mass enhancement due to coupling to phonons[7,11]. This comparison highlights the simplicity of the case of Bi$_2$O$_2$Se where any polaronic effect seems to be absent.

**Scaling between $A$ and $E_F$.** With the help of Fig. 3d, we can map $n$ to $E_F$ and translate the data of Fig. 2g in a new figure (Fig. 4a), which compares the evolution of $A$ with $E_F$ in Bi$_2$O$_2$Se and in SrTiO$_{3-\delta}$. Remarkably, the two sets of data join each other. Because of the lightness of its electrons, Bi$_2$O$_2$Se has a Fermi energy ten times higher than SrTiO$_{3-\delta}$ at the same carrier density.

In Fig. 4b, we include the new Bi$_2$O$_2$Se data in a universal plot of $A$ v.s. $E_F$. The data for other materials are taken from references[7,8,30] and are summarized in Supplementary Table 4–6. Note that for all anisotropic conductors including Bi$_2$O$_2$Se, the plot shows the prefactor in the plane with the higher conductivity.

This plot is an extension of the original Kadowaki–Woods plot[4] to dilute Fermi liquids. In a dense metal, the electronic-specific heat (in molar units) is an accurate measure of the Fermi energy. In a dilute metal, on the other hand, the molar units for atoms and electrons differ by orders of magnitude and therefore, one cannot use the electronic-specific heat as a measure of the Fermi energy.

Kadowaki and Woods observed originally that correlation between $A$ and $\gamma^2$ in heavy-electron metals, such $A/\gamma^2 \approx 10$ μΩ.

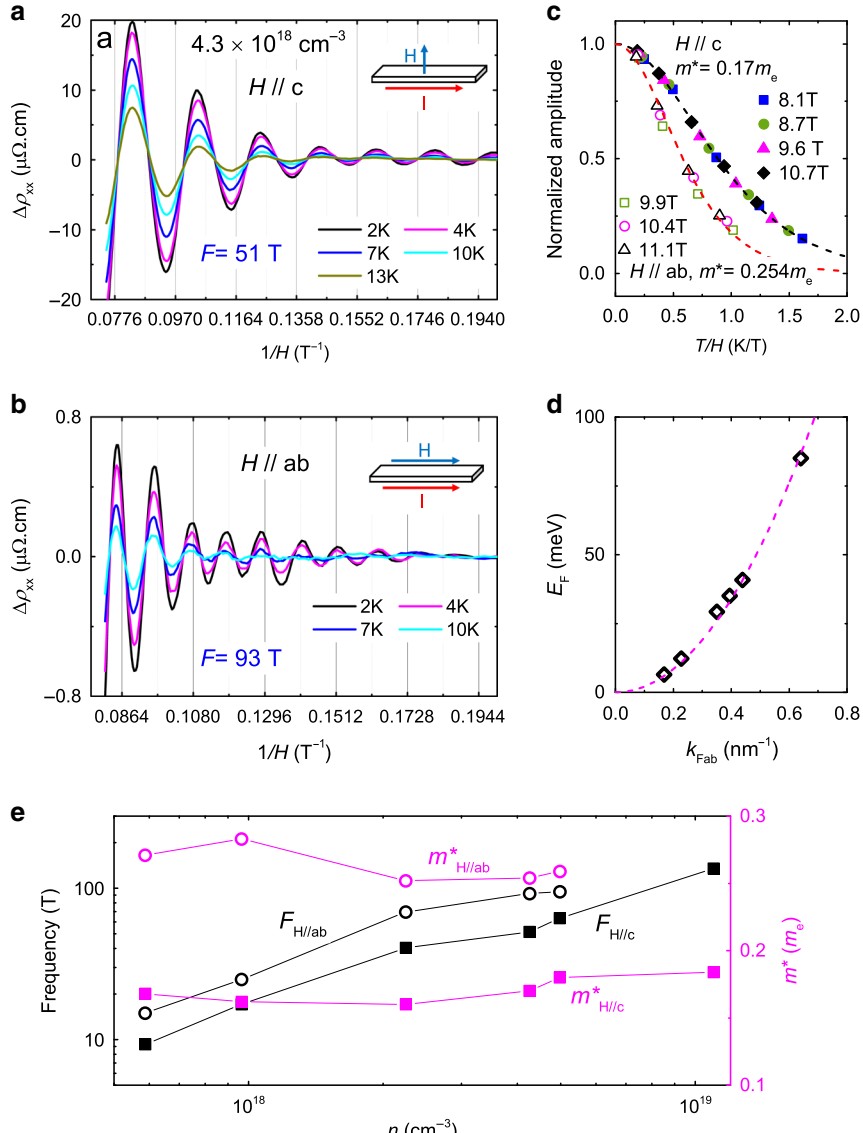

**Fig. 3 Shubnikov–de Haas effect for Bi$_2$O$_2$Se at $n \approx 4.3 \times 10^{18}$ cm$^{-3}$. a, b** Resistive quantum oscillations as a function of inverse field (1/$H$) with $H\|$ c and $H\|$ ab, respectively. $\Delta\rho = \rho - \rho_b$ where $\rho_b$ is a fitted background of the magneto-resistance. **c** Normalized amplitude of oscillations as a function of $T/H$ for two field directions. The dashed lines are the Lifshitz–Kosevich fits. **d** Fermi energy as a function of the in-plane Fermi wave vector for different samples, suggestive of a conducting band dispersion. The dashed line is a parabolic fit. **e** Evolution of the oscillation frequency ($F$) and the effective mass ($m^*$) with $n$ in different samples for field along c-axis and ab-plane, respectively. $F_{H\|c}$ and $F_{H\|ab}$ are denoted by solid black squares and open black circles, respectively. $m^*_{H\|c}$ and $m^*_{H\|ab}$ are denoted by solid magenta squares and open magenta circles, respectively.

cm.K$^2$.mol$^2$.J$^{-2}$[24] and contrasted it with a similar ratio noticed by Rice in elemental metals ($A/\gamma^2 \approx 0.4$ μΩ.cm.K$^2$.mol$^2$.J$^{-2}$)[3], where $\gamma$ is the electronic-specific heat coefficient. Subsequent studies[31] brought new data indicating that these ratios define two rough and lower boundaries (and recently even $^3$He has been shown to be on the Kadowaki–Woods plot[32]).

According to Fig. 4b, this 25-fold difference in the $A/\gamma^2$ magnitude is equivalent to a statement on the boundaries of $l_{quad}$, which lies between 1.6 and 40 nm across systems whose carrier concentration and Fermi energy differ by many orders of magnitude. This is in agreement with a recent observation by Kurita et al.[30]. They put under scrutiny the correlation between $A$ and the low-temperature slope of the Seebeck efficient (which remains a measure of the Fermi energy even in dilute systems[33]) and found that in a variety of systems $l_{quad} \approx 4$ nm[30].

$l_{quad}$ is a phenomenological quantity coming out of dimensional analysis. Nevertheless, it is well-defined and equal to the product of the Fermi wave vector and the cross-section of electron–electron collision[7,34]. This implies that its boundaries are meaningful and beg for an explanation.

## Discussion

Most previous theoretical attempts focus on isolated cases and did not seek a global scenario. Let us consider briefly their relevance to our data. One scenario for $T^2$ resistivity, proposed decades ago, invokes inelastic electron-impurity scattering and its interplay with electron–phonon interaction[35–37]. Such an effect has been reported in several impure metals. It appears too weak to account for the $T^2$ term in Bi$_2$O$_2$Se where $\frac{A}{\rho_0} \sim 10^{-3}$ K$^{-2}$. Pal and co-authors proposed that non-Galilean invariant FLs can display $T^2$

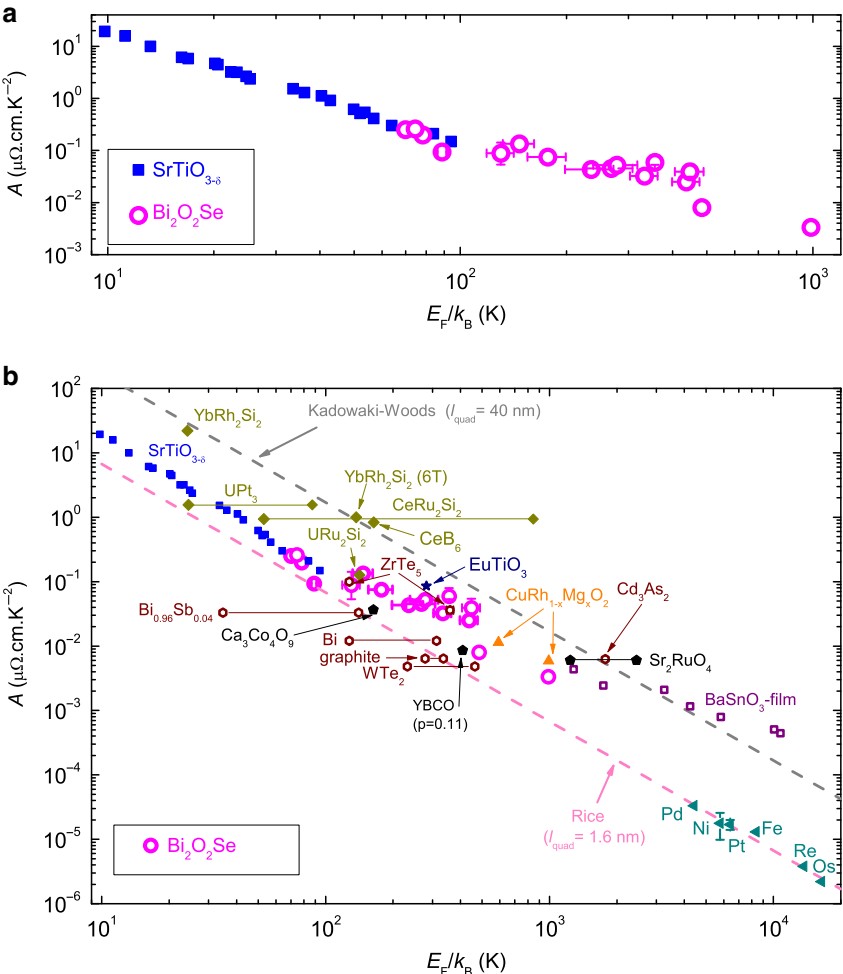

**Fig. 4 Universal scaling between the slope of $T^2$ resistivity ($A$) and Fermi energy ($E_F$). a** Variation of $A$ with $E_F$ on a Log–Log scale for $Bi_2O_2Se$: open magenta circles, compared to $SrTiO_3$: solid blue squares[7]. **b** $A - E_F$ plot across various FLs, such as Metals: solid dark cyan triangles; strongly correlated metals including heavy Fermions: solid dark yellow diamonds and YBCO ($YBa_2Cu_3O_y$), $Sr_2RuO_4$, $Ca_3Co_4O_9$: solid black pentagons; semimetals including Bi, $Bi_{0.96}Sb_{0.04}$, graphite, $WTe_2$, $Cd_3As_2$, and $ZrTe_5$: open wine hexagons; doped semiconductors including $Bi_2O_2Se$, $SrTiO_3$, $BaSnO_3$: open purple squares, $CuRhO_2$: solid orange triangles, $EuTiO_3$: solid navy star[7,8,30,46,47]. Most of the data are bounded by the two dashed lines set by Kadowaki–Woods and Rice, corresponding to a material-dependent length scale $l_{quad} \approx 40$ and 1.6 nm, respectively. The error bars for the data of $Bi_2O_2Se$ denote the uncertainty in determining $A$ and $E_F$ in processing the data. In solids with multiple Fermi surfaces, a horizontal bar links two data points representing the extrema in $E_F$.

resistivity even in the absence of Umklapp events[38]. However, this scenario for $T^2$ resistivity does not expect it in a systems with parabolic dispersion such as $Bi_2O_2Se$. Quantum interference near a ferromagnetic quantum critical point (QCP) can induce a resistivity proportional to $T^2 \ln T$[39]. This is inapplicable to $Bi_2O_2Se$, which is not close to any QCP. Lucas pointed to hydrodynamic flow of electrons in random magnetic fields as a possible source of $T^2$ resistivity[13] in $SrTiO_3$, speculating that oxygen vacancies can be magnetic there. Its relevance to a non-magnetic systems such as $Bi_2O_2Se$ is quite unlikely.

In summary, we find that the fermiology of dilute metallic $Bi_2O_2Se$ is such that interband or Umklapp scattering cannot happen. Nevertheless, there is a $T^2$ resistivity unambiguously caused by electron–electron scattering. We find a universal link between Fermi energy and the prefactor of $T$-square resistivity, which persists across various Fermi liquids. We conclude that a proper understanding of the microscopic origin of $T$-square resistivity in Fermi liquids is missing.

## Methods

**Sample growth**. $Bi_2O_2Se$ poly-crystals were synthesized through solid state reaction with stoichiometric Bi (5N), Se (5N), and $Bi_2O_3$ (5N) powders of high purity.

The mixed materials are sealed in an evacuated quartz tube and heated in an oven at 823K for 24 h. The single-crystalline phase was obtained through the chemical vapor transport (CVT) method by using poly-crystals as precursors. The sealed quartz tubes were placed in a horizontal furnace at a temperature gradient from 1123 to 1023 K over one week. The resulting single-crystals are shiny and air-stable. Note that, we didn't use transport agents such as $I_2$, in order to avoid unintentional doping. Samples with $n$ below and above $10^{18}$ cm$^{-3}$ are cleaved from two individual batches, respectively. Hence, we may expect a moderate inhomogeneous distribution of carrier concentrations in each sample.

**Experiments**. X-ray diffraction patterns were performed using a Bruker D8 Advanced X-ray diffractometer with Cu Kα radiation at room temperature. The composition of samples was determined by an energy-dispersive X-ray (EDX) spectrometer affiliated to a Zeiss field emission scanning electron microscope (SEM). The transport measurements were done by a standard four-terminal method in Quantum Design PPMS-Dynacool equipped with 14T magnet. Ohmic contacts were obtained by evaporating gold pad to samples before attaching wires with silver paste.

**DFT calculations**. All first-principles calculations were carried out using density functional theory (DFT) as implemented in Quantum Espresso[40]. The generalized gradient approximation (GGA) of Perdew–Burke–Ernzerhof revised for solids (PBEsol) type[41] was used to describe the exchange-correlation energy. We used ultrasoft pseudopotentials from the Garrity, Bennett, Rabe, Vanderbilt (GBRV) high-throughput pseudopotential set[42] and a plane-wave energy cutoff of 50 Ry and charge density cutoff of 250 Ry. The full phonon spectrum was calculated

using the supercell approach as implemented in phonopy[43] with the nonanalytical term correction at the Γ point included. As Bi is known for its strong spin-orbit coupling (SOC), the electronic band structure with SOC was calculated using the fully relativistic pseudopotentials taken from pslibrary (version 1.0.0)[44]. Both electronic band structure and phonon spectrum were calculated along the high-symmetry lines of the Brillouin zone of a body-centered tetragonal unit cell.

## Data availability

The data that support the findings of this study are included in this article and its supplementary information file and are available from the corresponding author upon reasonable request.

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

## Acknowledgements

This research was supported by National Natural Science Foundation of China via Project 11904294, Zhejiang Provincial Natural Science Foundation of China under Grant No. LQ19A040005 and the foundation of Westlake University. We thank the support provided by Dr. Chao Zhang from Instrumentation and Service Center for Physical Sciences (ISCPS) and computational resource provided by Supercomputer Center in Westlake University. This work is part of a DFG-ANR project funded by Agence Nationale de la Recherche (ANR-18-CE92-0020-01) and by the DFG through projects LO 818/6-1 and HE 3219/6-1.

## Author contributions

J.L.W. prepared the samples and did the experiments. He was assisted in the measurements by Z.K.X., J.F.W., W.H.H., and Z.R. J.W. and S.L. did DFT calculations. J.L.W., T.W., and X.L. prepared the figures. K.B. and X.L. wrote the manuscript. X.L. led the project. All authors contributed to the discussion.

## Competing interests

The authors declare no competing interests.
