## [Peer Review File · Nature Communications]

REVIEWER COMMENTS

Reviewer #1 (Remarks to the Author):

This manuscript reported a non-trivial phenomenon that the transport of dilute metallic Bi₂O₂Se displays T-square resistivity below the degeneracy temperature caused by electron-electron scattering, which neither Umklapp nor interband scattering are conceivable for inducing mechanism. Overall, the results are interesting and a set of control experiments was carried out to verify the non-trivial T-square resistivity proposed by the authors. However, the reviewer suspects the insufficient evidence cannot support this non-trivial phenomenon, at least in the present form, that can't meet the criterion of Nature Communications. The main issues are listed as following:

1) As claimed by the authors, the T-square resistivity without Umklapp and interband scattering in dilute metallic Bi₂O₂Se is caused by electron-electron scattering. Nevertheless, it lacks clear and sufficient evidence of transport data to further support the electron-electron scattering in dilute metallic Bi₂O₂Se. In addition, there are also need more experimental and theoretical evidence to demonstrate that the induced mechanism of the so-called non-trivial phenomenon is distinguished from Umklapp scattering.

2) The authors claimed the case of Bi₂O₂Se, a layered semiconductor with hard phonons, which becomes a dilute metal with a small single-component Fermi surface upon doping. As we know, there are unavoidable defects in as-synthesized Bi₂O₂Se crystals, such as Se, Bi, O Vacancies and Se-Bi, Bi-Se anti-site defects (Phys. Rev. B 2018, 97, 241203(R); Sci. Rep. 2018, 8, 10920.). The types of defects that led to formation of dilute metallic Bi₂O₂Se is not clear. It still needs further experimental or theoretical evidence to support the author's claim.

3) The authors used SrTiO₃ as a reference to explain the observed phenomenon. Nevertheless, it still lacks sufficient analysis of the difference of the band structure or crystal structure of both SrTiO₃ and Bi₂O₂Se to reveal the underlying mechanism which induced the T-square resistivity in dilute metallic Bi₂O₂Se.

4) The order of graphs in Figure 1 and 2 need to be corrected and the description of the graphs is confusing in manuscript. The type of fonts in Figure 1 is different.

Reviewer #2 (Remarks to the Author):

The article by Wang et al. report on the observation of a T^2 resistivity in ultra low carrier density samples of Bi₂O₂Se. As stated by the authors, this is remarkable because there is no obvious mechanism at such low carrier densities for the electron bath to relax its momentum during the presumed electron-electron scattering responsible for the T^2 temperature dependence. This is the second clear observation of such phenomena, with the first being reported by some of the same authors in SrTiO₃. I already find this fact on its own sufficient to justify publication in Nature Communications. However, the authors note that there are several unique features about Bi₂O₂Se, which provide new insight on trying to resolve this mystery. Due to the phonon spectrum, the lack of magnetic ions, and simple parabolic electronic dispersion, the authors are able to rule out three of the possible mechanisms, which were argued to have been possibly relevant in SrTiO₃. The manuscript is overall, very well written and deserves to be published in Nature Communication. I provide here a few additional comments, which I urge the authors to consider before publication.

- It was perhaps stated, but I certainly missed it if it was, that the carrier concentration reported in the paper was determined from the linear Hall resistivity measurements.

- At the end of page 4 the authors claim "it is straight forward to estimate the Fermi energy at a given carrier density." The authors need to explicitly state the expression that they use to convert the carrier density to a Fermi energy. Presumably it will be something like $E_F = \hbar^2/2m^*$

$(3\pi^2 n_{\text{Hall}})^{2/3}$. There is additional subtlety in presenting Fig. 3d as a result. By my estimate of the numbers I found that the authors appear to have used $m^* \sim 0.18 m_e$, or more likely - the effective mass from the SdH oscillation data. Again, the authors should be explicit as to which effective mass they used? Using the m^* from SdH would make sense though as the Fermi energy then reflects the property of the electrons near EF, which contribute to transport. However, Fig. 3d, presents a picture that looks like an electronic dispersion ($E(K)$) although it is correctly labeled as $EF(k)$). The quadratic dispersion that is drawn through the data further suggests such properties. If true, this would suggest that there is no electronic renormalization (for instance due to electron-phonon coupling). If true, that would be a stunning result. The reality is that I don't believe the authors can actually comment on the bare dispersion, since they only have the near EF effective mass. The parabolic dispersion is then a consequence of the constant effective mass as a function of carrier density. Unfortunately, I'm not sure how to rectify this figure, but the authors should state explicitly that the EF vs k plot is NOT a band dispersion $E(k)$ plot.

- On a related point, the authors might also note that the dispersion shown in Fig. 3d also appears consistent with the ARPES data which find $m^* = 0.14 m_e$. This perhaps allows for an independent determination of the dispersion, and if one takes the numbers seriously, then one actually can estimate a mass renormalization by the comparison of these two effective masses. Interestingly, the ARPES data do not seem capable of identifying a kink or e-phonon renormalization if it exists either. I don't know if there is a way to reconcile all these points.

- I'm curious if the authors have any thoughts as to why the deviation from the T^2 behavior is in the positive direction and nearly at the same temperature in most samples? Phonons are the one thing that occur to me to have a stronger than T^2 temperature dependence, but I would not naively expect the strength of the electron phonon scattering to change with electron density, but clearly the e-e scattering strongly depends on this. The deviation is very similar to that what was found for STO, and at a very similar temperature scale of ~ 30 K.

- My only other semi-significant complaint is that the authors don't address the potential role of transport anisotropy in the creation of Fig. 4a. From the ratio of the cyclotron effective masses, one can estimate that $m^*_c \sim 0.37 m_e$, which is twice the in-plane effective mass. Naively this would increase the A coefficient by a factor of 4. That doesn't change the order of magnitude estimates in fig. 4, but does question the matching in Fig. 4a, as a bit fortuitous. This is further emphasized by the fact that there is a clear break in slope between the STO data and the Bi2O2Se data, and the points where the data overlap are already in the multiband regime of STO, where the slope was argued to have already changed from single band expectations in ref. 7.

- The inset to figure 2g: how the residual resistivity depends on carrier density, is not mentioned in the figure caption or elsewhere. I am happy the data is included, and I presume the data is for Bi2O2Se, but it should be stated.

- I found the discussion in the SI ruling out other mechanisms very terse in general, and wouldn't mind if the authors expanded a bit on this. Most significantly, the authors rule out the quantum interference mechanism as "not relevant to the case here". Why is it not? An additional statement clarifying that would be useful.

- Is the dashed line in Fig. 1f,g the experimental lattice parameter, or simply the computational lattice parameter for ambient pressure?

Reviewer #3 (Remarks to the Author):

The authors report a comprehensive experimental characterization of the transport properties of the dilute Bi2O2Se for a broad range of temperatures and carrier concentrations.

These measurements reveal a quadratic dependence of the resistivity

on temperature.

The T-square dependence of the resistivity is a widely established behaviour in metals, whereby electron-electron scattering provides the dominant contribution to the resistivity.

The authors argue that Bi₂O₂Se may constitute an exceptional case for the observation of T-square resistivity, since the physical processes that are typically invoked to explain this phenomena, namely, Umklapp scattering and the coexistence of different electronic pockets, are excluded.

The authors discuss at length why this is the case and provide compelling experimental evidence that the Umklapp scattering may not take place, and that other electronic mechanisms can also be excluded. The authors further report the observation of a universal scaling law for the quadratic dependence on temperature.

The discovery of a new diluted metal exhibiting T-square resistivity may stimulate wide interest, and would certainly deserve publication in some forum. The claims and conclusions put forward in this study however are not entirely novel, since in Ref.[7] some of the authors report the observation of T-square resistivity for SrTiO₃ (STO). In that paper, similar arguments are put forward to illustrate that the T-square behaviour cannot be reconducted to the usual mechanisms (Umklapp and interband scattering). The novelty of this manuscript rely primarily on the difference between Bi₂O₂Se and STO. The authors argue that in Bi₂O₂Se, the electronic origin of the T-square behaviour is unambiguous, whereas in STO soft-phonons may be considered as a possible source of this behaviour. This discussion, however, is not entirely convincing, and is merely based on the phonon calculation reported in Fig. 1 (d-e) and (g).

To make this argument more solid the authors should truly rule out the emergence of soft acoustic modes, and discuss why they may underpin T-square resistivity.

Is this also due a Umklapp scattering process?

Can other modes acoustic modes soften upon small lattice distortion?

REVIEWER COMMENTS

Reviewer #1 (Remarks to the Author):

This manuscript reported a non-trivial phenomenon that the transport of dilute metallic Bi₂O₂Se displays T-square resistivity below the degeneracy temperature caused by electron-electron scattering, which neither Umklapp nor interband scattering is conceivable for inducing mechanism. Overall, the results are interesting and a set of control experiments was carried out to verify the non-trivial T-square resistivity proposed by the authors. However, the reviewer suspects the insufficient evidence cannot support this non-trivial phenomenon, at least in the present form, that can't meet the criterion of Nature Communications. The main issues are listed as following:

Our reply: We thank the referee for qualifying our results “interesting” and for recognizing the non-trivial aspect of the T-square resistivity. We are also grateful for the criticism. It helps us to clarify and sharpen our main message.

1) As claimed by the authors, the T-square resistivity without Umklapp and interband scattering in dilute metallic Bi₂O₂Se is caused by electron-electron scattering. Nevertheless, it lacks clear and sufficient evidence of transport data to further support the electron-electron scattering in dilute metallic Bi₂O₂Se.

Our reply: A Fermi liquid is expected to display T-square resistivity at low temperatures because of electron-electron scattering. Since the phase space for e-e scattering decreases quadratically with temperature, this would be the most natural candidate for explaining the origin of the one observed in Bi₂O₂Se. It is true that there are other (and more exotic) mechanisms for generating a T-square resistivity, which are unlikely to apply to Bi₂O₂Se. They were discussed in the supplement in the previous version. Following the referee's remarks, we have brought them to the main text in the revised version and argue that they are unlikely to be relevant to our observation.

In addition, there are also need more experimental and theoretical evidence to demonstrate that the induced mechanism of the so-called non-trivial phenomenon is distinguished from Umklapp scattering.

Our reply: Excluding the Umklapp scattering is straightforward. Since there is a small Fermi surface centered at the center of the Brillouin zone, collision between two cannot generate a wave-vector large enough to get out of the first Brillouin zone. The link between the size of the Fermi surface and Umklapp scattering is well-known. See Fig. 1 in ref.6 and Fig. 4 in ref.7. More specifically, an Umklapp event requires a Fermi wave vector larger than one-fourth of the smallest reciprocal lattice vector (G). For Bi₂O₂Se, the smallest G is along c-axis, $G_c = 2\pi/c \approx 5.17 \text{ nm}^{-1}$. Therefore, there is a threshold carrier density (n_U) for Umklapp scattering

$n = \frac{1}{3\pi^2} k_{Fc} k_{Fa}^2 = \frac{1}{3\alpha^2 \pi^2} k_{Fc}^3$ ($\alpha = \frac{k_{Fc}}{k_{Fa}}$). Given $k_{Fc} = \frac{G_c}{4}$ and $\alpha \approx 1.8$, n_U is deduced to

$3 \times 10^{19} \text{ cm}^{-3}$. The carrier density in samples of this study is well below n_U . We have added a more detailed discussion of this point in the new version.

2) The authors claimed the case of $\text{Bi}_2\text{O}_2\text{Se}$, a layered semiconductor with hard phonons, which becomes a dilute metal with a small single-component Fermi surface upon doping. As we know, there are unavoidable defects in as-synthesized $\text{Bi}_2\text{O}_2\text{Se}$ crystals, such as Se, Bi, O Vacancies and Se-Bi, Bi-Se anti-site defects (Phys. Rev. B 2018, 97, 241203(R); Sci. Rep. 2018, 8, 10920.). The types of defects that led to formation of dilute metallic $\text{Bi}_2\text{O}_2\text{Se}$ is not clear. It still needs further experimental or theoretical evidence to support the author's claim.

Our reply: We agree with the referee that there are unavoidable defects in as-grown $\text{Bi}_2\text{O}_2\text{Se}$ and the dopant leading to metallicity is not precisely identified. However, we do not understand the logic. The referee states: "The types of defects that led to formation of dilute metallic $\text{Bi}_2\text{O}_2\text{Se}$ is not clear". We fully agree with this, but do not understand why this sentence is immediately followed by this one? "It still needs further experimental or theoretical evidence to support the author's claim. Which "claim" depends on the identification of dopants? The existence of T-square resistivity? The absence of interband or Umklapp scattering? The agreement between the experimental and theoretical fermiology? To the best of our understanding, none of these issues is related to the identity of dopants.

Maybe the referee is surprised by this dilute metallicity driven by uncontrolled doping. However, the reason is well-known. Like many other semiconductors (PbTe and Bi_2Se_3 are prominent examples) a combination of high dielectric constant and low electron mass lowers the threshold for the metal-insulator transition. A rough estimation indicates that metallicity is expected for $\text{Bi}_2\text{O}_2\text{Se}$ when $n > 6 \times 10^{14} \text{ cm}^{-3}$. Therefore, dilute metallicity in samples with a carrier density of the order of 10^{18} cm^{-3} is unsurprising.

Maybe the referee is worried about the homogeneity of the metal. However, we detected quantum oscillations in resistivity which points to the existence of a single Fermi pocket as expected from band structure calculations. For samples with $n > 10^{18} \text{ cm}^{-3}$, the carrier density deduced from Hall effect and quantum oscillation are close to each other (see Table S1). This indicates that there is a uniform metallic phase in these samples.

Therefore, while it is true that we do not know which defects are making the system metallic, none of our conclusions are affected by the absence of this knowledge.

3) The authors used SrTiO_3 as a reference to explain the observed phenomenon. Nevertheless, it still lacks sufficient analysis of the difference of the band structure or crystal structure of both SrTiO_3 and $\text{Bi}_2\text{O}_2\text{Se}$ to reveal the underlying mechanism which induced the T-square resistivity in dilute metallic $\text{Bi}_2\text{O}_2\text{Se}$.

Our reply: In order to address this point, we have added some additional information on SrTiO_3 in the revised version.

SrTiO₃: a. Crystal structure

b. Conducting band

c. Fermi surface when only the lowest band is occupied

The conducting band and the Fermi surface of SrTiO₃ are shown above. We summarize the difference and similarity between Bi₂O₂Se and SrTiO₃ in the table below, which has been included in the revised supplement. Bi₂O₂Se and SrTiO₃ both host a Fermi pocket at the Γ point. In both T-square resistivity survives when neither inter-band nor Umklapp scattering is allowed. The crystal structure and the phonon spectrum are different in the two systems, but the prefactor of T-square resistivity scales with the Fermi energy similarly.

Our main result provides a well-defined theoretical challenge to our community: What mechanism leads to T-square resistivity with a scalable prefactor in dilute metals no matter their crystal structure or phonon spectrum?

	Crystal	Band position	Band dispersion	Fermi pocket
Bi ₂ O ₂ Se	layered tetragonal Anti-ThCr ₂ Si ₂ (I4/mmm)	Γ point	parabolic	ellipsoid
SrTiO ₃	3D cubic Perovskite (Pm-3m)	Γ point	non-parabolic	squeezed ellipsoid

4) The order of graphs in Figure 1 and 2 need to be corrected and the description of the graphs is confusing in manuscript. The type of fonts in Figure 1 is different.

Our reply: Thanks for pointing out this. This has been corrected in the revised version.

Reviewer #2 (Remarks to the Author):

The article by Wang et al. report on the observation of a T² resistivity in ultra low carrier density samples of Bi₂O₂Se. As stated by the authors, this is remarkable because there is no obvious mechanism at such low carrier densities for the electron bath to relax it's momentum during the presumed electron-electron scattering responsible for the T² temperature dependence. This is the second clear observation of such phenomena, with the first being reported by some of the same authors in SrTiO₃. I already find this fact on its own sufficient to justify publication in Nature Communications. However, the authors note that there are several unique features about Bi₂O₂Se, which provide new insight on trying to resolve this mystery. Due

to the phonon spectrum, the lack of magnetic ions, and simple parabolic electronic dispersion, the authors are able to rule out three of the possible mechanisms, which were argued to have been possibly relevant in SrTiO₃. The manuscript is overall, very well written and deserves to be published in Nature Communication. I provide here a few additional comments, which I urge the authors to consider before publication.

Our reply: We thank the referee for this positive evaluation and the insightful remarks.

1) It was perhaps stated, but I certainly missed it if it was, that the carrier concentration reported in the paper was determined from the linear Hall resistivity measurements.

Our reply: Thanks for pointing to this shortcoming. The carrier concentration (n) was indeed determined from the Hall effect. This has been explicitly mentioned in the revised version.

2) At the end of page 4 the authors claim “it is straight forward to estimate the Fermi energy at a given carrier density.” The authors need to explicitly state the expression that they use to convert the carrier density to a Fermi energy. Presumably it will be something like $E_F = \hbar^2/2m^* (3\pi^2 n_{\text{Hall}})^{2/3}$. There is additional subtlety in presenting Fig. 3d as a result. By my estimate of the numbers I found that the authors appear to have used $m^* \sim 0.18 m_e$, or more likely – the effective mass from the SdH oscillation data. Again, the authors should be explicit as to which effective mass they used?

Our reply: The referee is right. We indeed used the SdH mass. As one can see in table S1, the mass does not evolve much with increasing doping and is within the range of 0.16 to 0.18 m_e . We have made this clear in the revised version that the Fermi energy and the Fermi wave-vector of Fig. 3d have been estimated for each sample has been extracted from the frequency and the mass obtained in SdH measurements.

3) Using the m^* from SdH would makes sense though as the Fermi energy then reflects the property of the electrons near E_F , which contribute to transport. However, Fig. 3d, presents a picture that looks like an electronic dispersion ($E(K)$ although it is correctly labeled as $E_F(k)$). The quadratic dispersion that is drawn through the data further suggests such properties. If true, this would suggest that there is no electronic renormalization (for instance due to electron-phonon coupling). If true, that would be a stunning result. The reality is that I don't believe the authors can actually comment on the bare dispersion, since they only have the near E_F effective mass. The parabolic dispersion is then a consequence of the constant effective mass as a function of carrier density. Unfortunately, I'm not sure how to rectify this figure, but the authors should state explicitly that the E_F vs k plot is NOT a band dispersion $E(k)$ plot.

Our reply: The referee is right. Our experimental data does not lead to any statement on the bare dispersion. The measured SdH mass is almost constant. The measured oscillation frequency yields the square of the Fermi wave-vector. Therefore, the parabolic dispersion seen in Fig. 3d is a direct consequence of constant m^* . Let us highlight that this is not always the case. In the case of SrTiO₃, the non-parabolic dispersion of the lower band shows itself in an

evolving SdH mass as seen in the figure below. Therefore, using the same procedure, one finds that E_F is not quadratic in k_F . What is seen in the figure represents $E_F(k_F)$. Only in a rigid band picture this is equivalent to the dispersion $E(k)$.

4) On a related point, the authors might also note that the dispersion shown in Fig. 3d also appears consistent with the ARPES data which find $m^* = 0.14 m_e$. This perhaps allows for an independent determination of the dispersion, and if one takes the numbers seriously, then one actually can estimate a mass renormalization by the comparison of these two effective masses. Interestingly, the ARPES data do not seem capable of identifying a kink or el-phonon renormalization if it exists either. I don't know if there is a way to reconcile all these points.

Our reply: We thank the referee for this insightful remark, which had escaped our attention. Indeed, the SdH mass in $\text{Bi}_2\text{O}_2\text{Se}$ is $0.16\text{--}0.18m_e$. This is close to the ARPES mass ($0.14m_e$) and suggests experimental consistency. In the new version, we compare these numbers with the DFT calculated band mass which is $0.125m_e$. Thus, the SdH mass is only 1.3–1.5 times larger than the bare mass. This modest mass renormalization is to be contrasted with the case of STO where the cyclotron mass is 2.5 times the bare mass. This is yet another argument in favor of weak polaronic effects in the system under study.

5) I'm curious if the authors have any thoughts as to why the deviation from the T^2 behavior is in the positive direction and nearly at the same temperature in most samples? Phonons are the one thing that occur to me to have a stronger than T^2 temperature dependence, but I would not naively expect the strength of the electron phonon scattering to change with electron density, but clearly the e-e scattering strongly depends on this. The deviation is very similar to that what was found for STO, and at a very similar temperature scale of ~ 30 K.

Our reply: The referee raises an interesting point. In SrTiO_3 , the ferroelectric soft phonons are suspected to play a decisive role. In $\text{Bi}_2\text{O}_2\text{Se}$, phonon scattering should also play a role. The figure below compares the evolution of the deviation temperature (T_{quad}) with carrier density in $\text{Bi}_2\text{O}_2\text{Se}$ and SrTiO_3 . One can see that it is much steeper in the latter. The referee is right that the strength of electron-phonon scattering is not expected to vary with doping. However, the temperature above which the electrons become non-degenerate does. Now, the scattering cross section for electron-phonon collisions is not the same for degenerate and non-degenerate electrons. This may provide a reason for the evolution of T_{quad} with n .

6) My only other semi-significant complaint is that the authors don't address the potential role of transport anisotropy in the creation of Fig. 4a. From the ratio of the cyclotron effective masses, one can estimate that $m^*_c \sim 0.37 m_e$, which is twice the in-plane effective mass. Naively this would increase the A coefficient by a factor of 4. That doesn't change the order of magnitude estimates in fig. 4, but does question the matching in Fig. 4a, as a bit fortuitous. This is further emphasized by the fact that there is a clear break in slope between the STO data and the $\text{Bi}_2\text{O}_2\text{Se}$ data, and the points where the data overlap are already in the multiband regime of STO, where the slope was argued to have already changed from single band expectations in ref. 7.

Our reply: Unfortunately, the size of most samples in this study is around 1 millimeter. Therefore, measuring the out-of-plane resistivity in these samples is not easy. We will try to grow larger samples available for out-of-plane resistivity in future. It is true that indeed that the out-of-plane T-square resistivity prefactor is expected to be larger. In contrast to STO, $\text{Bi}_2\text{O}_2\text{Se}$ is anisotropic. Therefore, the perfect overlap is indeed fortuitous. However, let us recall that for all anisotropic conductors (including graphite, YBCO and Sr_2RuO_4) in Fig. 4b, it is the in-plane prefactor, which is plotted. It is the same for $\text{Bi}_2\text{O}_2\text{Se}$.

7) The inset to figure 2g: how the residual resistivity depends on carrier density, is not mentioned in the figure caption or elsewhere. I am happy the data is included, and I presume the data is for $\text{Bi}_2\text{O}_2\text{Se}$, but it should be stated.

Our reply: Thank you for this. The inset of Fig. 2g is mentioned in the revised version.

8) I found the discussion in the SI ruling out other mechanisms very terse in general, and wouldn't mind if the authors expanded a bit on this. Most significantly, the authors rule out the quantum interference mechanism as "not relevant to the case here". Why is it not? An additional statement clarifying that would be useful.

Our reply: Following referee's suggestion, we have moved the discussion of other possible mechanisms from the supplement to the main text in the revised version. We have added a couple of sentences to discuss the possible relevance of quantum-interference mechanism.

9) Is the dashed line in Fig. 1f,g the experimental lattice parameter, or simply the computational lattice parameter for ambient pressure?

Our reply: The dashed line marks the optimized computational lattice parameter at ambient pressure, which is 3.873 Å. The experimental lattice parameter is 3.88 Å. We have stated this in the caption of Fig. 1 of the revised version.

Reviewer #3 (Remarks to the Author):

The authors report a comprehensive experimental characterization of the transport properties of the dilute Bi₂O₂Se for a broad range of temperatures and carrier concentrations.

These measurements reveal a quadratic dependence of the resistivity on temperature.

The T-square dependence of the resistivity is a widely established behavior in metals, whereby electron-electron scattering provides the dominant contribution to the resistivity.

The authors argue that Bi₂O₂Se may constitute an exceptional case for the observation of T-square resistivity, since the physical processes that are typically invoked to explain this phenomena, namely, Umklapp scattering and the coexistence of different electronic pockets, are excluded. The authors discuss at length why this is the case and provide compelling experimental evidence that the Umklapp scattering may not take place, and that other electronic mechanisms can also be excluded. The authors further report the observation of a universal scaling law for the quadratic dependence on temperature.

The discovery of a new diluted metal exhibiting T-square resistivity may stimulate wide interest, and would certainly deserve publication in some forum. The claims and conclusions put forward in this study however are not entirely novel, since in Ref.[7] some of the authors report the observation of T-square resistivity for SrTiO₃ (STO). In that paper, similar arguments are put forward to illustrate that the T-square behaviour cannot be reconducted to the usual mechanisms (Umklapp and interband scattering). The novelty of this manuscript rely primarily on the difference between Bi₂O₂Se and STO. The authors argue that in Bi₂O₂Se, the electronic origin of the T-square behaviour is unambiguous, whereas in STO soft-phonons may be considered as a possible source of this behaviour. This discussion, however, is not entirely convincing, and is merely based on the phonon calculation reported in Fig. 1 (d-e) and (g).

Our reply: We thank the referee for this fine summary of our result and its context. It is true that the two conclusions of this paper are very similar to the conclusions of reference 7: i) T-square resistivity can occur even in absence of the two known mechanisms for generating it from e-e scattering. ii) The magnitude of the prefactor of T-square resistivity in a Fermi liquid can be predicted knowing its Fermi energy.

Both these messages were already presented in reference 7. It is therefore legitimate to wonder what is new and how significant are a similar conclusion with Bi2O2Se data.

Here are our main arguments:

First, the discovery of a second system implies that the case of SrTiO₃ is not unique. Before the publication of reference 7 in 2015, it was taken for granted that T-square resistivity in STO is due to e-e scattering. When it became clear that this T-square resistivity survives in the extreme dilute limit without either Umklapp or interband scattering, skeptics pointed their finger to soft phonons and suggested that they may cause T-square resistivity. After all, STO is a strange solid. This second system makes our observation more generic.

Second, the observation is cleaner. The T-square resistivity is restricted to temperature well below the degeneracy temperature.

Third, the solid in question is much simpler. Not only, it has no imaginary phonon modes. Its band structure is parabolic. There is only one single band and the mass renormalization is negligible.

Fourth and most importantly, the magnitude of the T-square prefactor scales with the Fermi energy as expected. This implies that what sets the prefactor is NOT the carrier concentration or the effective mass, but their combination, which yields the Fermi energy.

For all these reasons, the present results are significant in highlighting the absence of a satisfactory explanation of the ubiquity of T-square resistivity in Fermi liquids.

1) To make this argument more solid the authors should truly rule out the emergence of soft acoustic modes, and discuss why they may underpin T-square resistivity.

Our reply: We are not aware of any scattering mechanism involving soft acoustic phonons. On the other hand, a possible mechanism involving soft transverse optical phonons (TO) was proposed by Epifanov ref[15]. In his picture, coupling between electrons and a ferroelectric soft phonon coupling leads to the following expression for the relaxation time

$$\tau(E) = \frac{128\pi^3 \hbar^3 \delta^2}{g^2 m k_B^2 \varphi(l) T^2}$$

Here; \hbar is the reduced Planck constant, k_B is the Boltzmann constant, m is the effective mass of electrons, g is an electron-phonon interaction constant, δ is a constant reflecting the wave number dependence of permittivity $\varepsilon(q) = [\varepsilon_0^{-1} + \delta q^2]^{-1}$, ε_0 is static permittivity, q is the phonon wave number and $\varphi(l)$ is a complex power-law formula with $l = 2mE\varepsilon_0\delta\hbar^{-2}$ and E the kinetic energy.

Now, since SrTiO_3 is quantum paraelectric, such a scenario cannot be excluded there. On the other hand, $\text{Bi}_2\text{O}_2\text{Se}$ is not a quantum paraelectric and there is no reason to suspect the relevance of this scenario.

2) Is this also due a Umklapp scattering process?

Our reply: We discussed this in our response to referee 1. An Umklapp even requires a sufficiently large Fermi surface., which is not the case of the system under study. The Fermi surface is a small pocket centered at the center of the Brillouin zone. In this context, all electronic wave-vectors are too small to generate a wave-vector large enough to get out of the first Brillouin zone. It is commonly believed that a Fermi surface of sufficiently large radius is required for Umklapp. This is not the case of our system. We have added a detailed discussion of the Umklapp conditions in the new version.

3) Can other modes acoustic modes soften upon small lattice distortion?

Our reply: According to the calculated phonon dispersion at different pressures, the answer is negative. The figures below show the phonon dispersion at pressures of 16GPa, 0GPa and -2GPa. We find that the lowest optical (TO1) mode (marked in the figure) softens more dramatically than other modes by reducing the pressure, which is consistent with Fig. 1f. The acoustic modes show some softening too, but their evolution is much less dramatic.

REVIEWERS' COMMENTS:

Reviewer #1 (Remarks to the Author):

The authors use layered Bi₂O₂Se as a platform to investigate the origin of T-square resistivity in Fermi liquids. It seems to us that this problem is universal and sufficiently important for various material systems. However, there are still some issues to be addressed.

1. In fig.2b-e, all the data points deviate from T-square dependence at 20 or 30 K. Please fit the relationship more precisely, i.e. $R \propto T^{2.2}$, and analyze to what extent will this deviation affect your conclusions.
2. Please show some details about the transport measurements. What is the device structure? How do you apply such a strong magnetic field?
3. In another paper related to Bi₂O₂Se (Wu et al. Nano lett.2017), the authors show that positive pressure can lead to ferroelectricity, while your results seems to be opposite. Could you comment on this paper?
4. Even if it is a guess, could you please propose a possible mechanism for T-square resistivity dependence of Bi₂O₂Se?

Reviewer #2 (Remarks to the Author):

In my opinion, the resubmission by Wang et al. has satisfactorily addressed all the points raised by the referees previously. My only complaint is that Fig. 3d is still referred to as a band dispersion, which is technically incorrect. I would prefer if the authors removed the clause "mimic the conducting band dispersion" in the figure caption. Possibly "suggestive of a conduction band dispersion" might be better? More egregious is the sentence in the text that states "As a consequence, the resolved dispersion is parabolic." Perhaps the authors could state "As a consequence, the data appears to suggest a parabolic dispersion, but we note that this picture does not capture the potential undressing of the electrons at higher binding energies."

Reviewer #3 (Remarks to the Author):

The authors have addressed all my questions exhaustively and convincingly. I recommend the manuscript for publication in Nature Communications.

REVIEWERS' COMMENTS:

Reviewer #1 (Remarks to the Author):

The authors use layered Bi2O2Se as a platform to investigate the origin of T-square resistivity in Fermi liquids. It seems to us that this problem is universal and sufficiently important for various material systems. However, there are still some issues to be addressed.

Our reply: We thank the referee for the fair evaluation of our results and the additional comments. Our point-by-point replies are listed below.

1. In fig.2b-e, all the data points deviate from T-square dependence at 20 or 30 K. Please fit the relationship more precisely, i.e. $R \propto T^{2.2}$, and analyze to what extent will this deviation affect your conclusions.

Our reply: The linear behavior at low temperatures in Fig. 2b-e clearly indicates the T-square resistivity in our samples, that is a manifestation of Fermi Liquids. We do not think our conclusion will be affected by choosing different kinds of fitting. Concerning the requirement by the referee, we refitted our data by the following power-law equation $\rho = \rho_0 + AT^\alpha$. We found that α is very close to 2 in our samples and T_{quad} lies around 30K. For clarity, we show the fitting figures for several samples below.

2. Please show some details about the transport measurements. What is the device structure? How do you apply such a strong magnetic field?

Our reply: The measurements were done on Bi₂O₂Se single crystals of millimeter size, for which Ohmic contacts were obtained by evaporating gold pads to samples before attaching wires with silver paste. The picture for a sample with gold pads was shown in Fig. S2b. The transport measurements were done by a standard four-terminal method in Quantum Design PPMS. All this information was listed in

“Method” of the previous version. The magnetoresistance was measured up to 14T in PPMS. This information has been added in “Method” in the revised version.

3. In another paper related to Bi₂O₂Se (Wu et al. Nano lett.2017), the authors show that positive pressure can lead to ferroelectricity, while your results seems to be opposite. Could you comment on this paper?

Our reply: Thanks for sharing with us this theoretical paper. We find that our calculation is compatible with, rather than opposite to it. We know from this paper that Bi₂O₂Se is expected to exhibit ferroelectricity upon uniaxial or biaxial extensive strain. The critical biaxial strain is around 1.7%. Similarly, according to our calculation, the TO₁ phonon freezes upon negative hydrostatic pressure P_c~-5GPa, corresponding to an extension of lattice parameter (a) by 2.3%.

4. Even if it is a guess, could you please propose a possible mechanism for T-square resistivity dependence of Bi₂O₂Se?

Our reply: To be honest, we have no idea about the mechanism. Below is just a conjecture. The T-square resistivity may arise from electron-electron scattering assisted by other quasiparticles, such as phonons. This conjecture should be carefully checked by sophisticated theoretical calculations and compared with the experiments.

Reviewer #2 (Remarks to the Author):

In my opinion, the resubmission by Wang et al. has satisfactorily addressed all the points raised by the referees previously. My only complaint is that Fig. 3d is still referred to as a band dispersion, which is technically incorrect. I would prefer if the authors removed the clause “mimic the conducting band dispersion” in the figure caption. Possibly “suggestive of a conduction band dispersion” might be better? More egregious is the sentence in the text that states “As a consequence, the resolved dispersion is parabolic.” Perhaps the authors could state “As a consequence, the data appears to suggest a parabolic dispersion, but we note that this picture does not capture the potential undressing of the electrons at higher binding energies.”

Our reply: We are grateful that the referee was satisfied by our reply and sorry for that the previous description about the band dispersion is not as clear as expected by the referee. In the revised version, the phrase “mimic the conducting band dispersion” has been replaced by “suggestive of a conducting band dispersion” according to the referee’s suggestion. The sentence “As a consequence, the resolved dispersion is parabolic.” has been replaced by “The data imply that the conducting band dispersion is parabolic.”